# OpenReview forum: "A Coin Flip for Safety: LLM Judges Fail to Reliably Measure Adversarial Robustness"
_ICML.cc/2026/Conference — ICML 2026 regular_

### Official Review · Reviewer_wwTo · 2026-03-04

**Soundness:** 2
**Presentation:** 3
**Significance:** 3
**Originality:** 4
**Overall Recommendation:** 4
**Confidence:** 3

**Summary:**

This paper studies the reliability of LLM-as-a-Judge systems for adversarial safety evaluation. The authors conduct a large-scale empirical audit with human annotations, covering multiple behaviors, victim models, attack methods, and judge models. They show that under distribution shift inherent to read-teaming, judge performance can degrade substantially and even approach random guess, inflating the commonly reported attack success rates. As a consequence, the ASR may reflect evaluator failure rather than genuinely harmful model behavior. Building on these findings, the authors introduces two benchmarks, ReliableBench and JudgeStressTest, for a more reliable measurement of adversarial robustness.

**Compliance With Llm Reviewing Policy:**

Affirmed.

**Final Justification:**

Many of my concerns are addressed, especially by the additional annotation. So I raised my score accordingly.

**Key Questions For Authors:**

See Weaknesses.

**Limitations:**

See Weaknesses.

**Strengths And Weaknesses:**

**Strengths**

- The paper targets a critical challenge in current safety evaluation. While large-scale red-teaming has made LLM-as-a-Judge a standard for measuring adversarial robustness, its reliability hasn't been verified.
- The work provides a large human-labeled audit with a clear and fine-grained rubric and quality control. This constitutes a valuable resource for the community and strengthens the credibility of the empirical claims
- The experiment covers multiple behaviors, victim models, attack methods, and judge models. By stratified analyses, the authors decompose distribution shift into attack shift, model shift and data shift, and identify when and how judge performance degrades and how these factors interact. The analyses on ROC and training dynamics also support that the performance degradation is not due to thresholding, and is largely exploiting reward noise instead of amplifying genuinely harmful outputs.
- The clear drop and spread across behaviors and categories in Figure 4 illustrate heterogeneity, supporting the claim that some behaviors are unsuitable for reliable benchmarking. The release of ReliableBench and JudgeStressTest further help to facilitate more reliable evaluation protocols in future work.

**Weaknesses**

- Although restricting human annotation to StrongREJECT-positive samples is understandable due to annotation cost, this design introduces a potential selection bias as many of the samples may concentrate near the decision boundary of that filter. This makes it difficult to disentangle whether the observed failure arise from limitations of filter itself, the LLM judges, or the systematic false discovery that is unavoidable for all classification algorithms.
- The corrected ASR appears somewhat heuristic and lacks a clear statistical justification. Precision may depend on the attack method and harm categories. Using a global estimation of precision to adjust ASR may introduce distributional dependence to the annotation dataset and reduce robustness across harm distribution. Also, the precision is estimated only on samples flagged by StrongREJECT, which may introduce selection bias and underestimate the precision. It would be helpful if the authors could clarify the exact formula and the assumptions underlying this correction method.
- ReliableBench is constructed by selecting behaviors with high judge-human agreement, which may bias the benchmark toward easier cases. As a result, evaluation results based on RB may overestimate the model’s robustness. Also, the RB and JST datasets depend on the capability of current judge models. How should we deal with the improvement of judge models correspondingly?
- L394 and L426 appear to report inconsistent percentages. This might be a typo.
- The paper use “expected ASR” and “corrected ASR” simultaneously. Are they refer to the same quantity?

---

> ### Author Rebuttal · Authors · 2026-03-29
>
> Thanks for the thoughtful review and attention to detail!
>
> **W1: Prefiltering with StrongREJECT introduces selection bias, making it difficult to attribute failures to the judges or the filter.**
>
> **A1:** We agree that unbiased sampling would be preferable given an unlimited labeling budget and that quantitative analysis on negatives would provide a more comprehensive picture.
>
> To address this concern, we annotated a new set of 20 negatively judged generations per attack across all models (400 samples in total) using the same procedure as in the original paper. We found that only 13% were false negatives. This now allows us to also analyze judge reliability on negative samples and make statements about recall, providing a more complete view of judge performance. At the same time, the low false-negative rate confirms that the information yield per label under unbiased sampling would be extremely low compared to our prefiltering approach, validating our original design choice and providing concrete evidence for it.
>
> **Action:** We thank the reviewer for this comment and added this quantitative analysis of negatively judged samples to the paper, enabling analysis of both positive and negative judgments and better supporting our labeling strategy. For results, see Figure 3 (https://figshare.com/s/0e60460a72506d06ca65?file=63267886).
>
> **W2: Corrected ASR is heuristic; precision varies by attack and harm category; global estimation may be imprecise and is affected by SR-filtering bias. Please clarify the formula and underlying assumptions.**
>
> **A2:** We appreciate the attention to detail!. The corrected (now always referred to as expected) ASR is defined as:
>
> $ASR_{expected} = ASR_{reported} \times \left(1 - (1 - precision)^N\right)$
>
> where $N$ describes the number of observed judged positive samples, and precision is estimated on the subset of judge-positive samples we annotated. Intuitively, this describes the probability of encountering at least one true positive (or more) in a set of N positively judged samples. Importantly, we estimated this for every combination of judge, victim model, and attack when calculating the expected ASR. Ideally, this estimation would be performed for each individual behavior, but our annotation budget did not support sufficiently accurate per-behavior estimates. Precision is not assumed to be constant across victim models or attacks, and we agree that relying on a single global estimate would be unreasonable.
>
> **Action** We added the formula for calculating the expected ASR to our paper, along with a description of the underlying assumptions (e.g., that precision is estimated separately for every combination of victim model and attack).
>
> **W3: ReliableBench may be biased toward easier cases; JudgeStressTest depends on the capability of current judge models. How should we deal with improving judge models over time?**
>
> **A3:** These are important design questions, and we welcome the opportunity to clarify the intended role of our benchmarks.
>
> *ReliableBench* is a complement, not a replacement, for full-dataset evaluation. We do not propose that researchers abandon evaluation on the full HarmBench distribution. ReliableBench is practically free by design: since it is a strict subset of HarmBench behaviors, computing results on it requires no additional work. It provides an additional, less noisy estimate of harmfulness. If results on ReliableBench and the full dataset diverge substantially, this signals researchers to investigate why their judge struggles on certain behaviors.
>
> *JudgeStressTest* is explicitly designed to track judge progress over time. It allows researchers to directly evaluate judge performance on difficult-to-judge samples and measure improvement as new judges are developed. The two benchmarks complement each other: if a judge performs poorly on JudgeStressTest, researchers know it struggles with hard cases and should rely more on ReliableBench, where judge-human agreement is higher. Conversely, strong performance on JudgeStressTest gives researchers greater confidence in results on the full dataset. JudgeStressTest thus directly addresses the concern about evolving judge capabilities by providing the mechanism to assess whether a judge is reliable enough beyond the easier cases captured by ReliableBench.
>
> **Action** We thank the reviewer for this comment and will add this description of the role of both datasets to the paper.
>
> **W4: Typo regarding percentages.**
>
> Thank you for the careful reading; this was indeed a typo and has been corrected. We reviewed the paper in detail once more and found another typo in line 241, which we also corrected.
>
> **W5: "Corrected ASR" vs. "Expected ASR": are these the same?**
>
> Yes, they refer to the same quantity. We have standardized the wording throughout the paper to use "Expected ASR," as it more accurately reflects the metric (an approximation rather than a definitive correction).

---

> > ### Author Rebuttal · Reviewer_wwTo · 2026-04-04
> >
> > Thanks for the clarifications. Many of my concerns are addressed, especially by the additional annotation. So I raised my score accordingly.

---

> > > ### Author Response · Authors · 2026-04-07
> > >
> > > We appreciate the thoughtful feedback! We are happy that we were able to resolve most concerns.

---

### Official Review · Reviewer_p2t1 · 2026-03-10

**Soundness:** 3
**Presentation:** 3
**Significance:** 3
**Originality:** 2
**Overall Recommendation:** 4
**Confidence:** 2

**Summary:**

This paper shows that existing LLM judges are not reliable in adversarial robustness evaluation. To improve the reliability of LLM judges, the authors propose two benchmarks (i.e., ReliableBench and JudgeStressTest) and a judge concordance score. ReliableBench contains easy-to-judge samples, aiming to mitigate the evaluation bias of LLM judges. JudgeStressTest contains failure cases of LLM judges, which is used for investigating why LLM judges fail. The judge concordance score measures the degree of consensus among N judges for a given prompt-generation pair, serving as a diagnostic tool to identify behaviours for which judges share common logic.

**Compliance With Llm Reviewing Policy:**

Affirmed.

**Final Justification:**

My concerns have been addressed, so I will increase my score to 4.

**Key Questions For Authors:**

Please refer to my major weaknesses.

**Limitations:**

Yes.

**Strengths And Weaknesses:**

**Strengths**:

1. Robust evaluation is a pain point in adversarial domain. This paper provides sufficient insights on LLM judges in adversarial settings, which may inspire more researchers in this domain to rethink robust evaluation.
2. This paper conducts comprehensive experiments to justify its claims.

**Weaknesses**:

major:

1. Lack of evaluation against more advanced victim models. The authors are encouraged to include more advanced open-source victim models such as GPT-oss-20b, 120b and GPT-oss-safeguard-20b, 120b.
2. A follow-up weakness is that it remains unclear whether the takeaways would generalize to closed-source models. Although the authors explain why proprietary models are excluded, the claim ‘LLM judges fail to reliably measure adversarial robustness’ seems insufficiently supported if more advanced commercial models such as GPT, Gemini and Claude are excluded. The author could either claim ‘Open-source LLM Judges fails to …’ or include more closed-source models into the experiments.

minor:

1. line 47, the left quotation mark is incorrect.

I am not an expert in LLM-as-a-judge field, so I will open to other reviewers’ opinions and I will be willing to increase the score if my concerns can be well-addressed.

---

> ### Author Rebuttal · Authors · 2026-03-29
>
> We thank the reviewer for their helpful comments that helped us to improve our paper. New results can be found here: https://figshare.com/s/0e60460a72506d06ca65
>
> **W1: Lack of evaluation against more sophisticated victim models.**
>
> **A1:** We agree that results on more capable models would strengthen our claims. Labeling is labor-intensive, and we had to allocate our annotation budget carefully, prioritizing models of varying size that are frequently used as victim models in the adversarial ML community. We believe this focus is well-motivated, as judge reliability matters most for the models researchers are actually evaluating in practice. Nevertheless, we agree that results on capeable and new models are of interest and conducted new experiments using GPT-OSS 120B as a victim model, labeling 50 samples per attack using the same protocol as the rest of our study. Due to the constrained rebuttal timeline, we were not able to label a larger sample and will conduct further labeling for the camera-ready version.
>
> The results largely follow those of other models we tested, where larger models are easier to judge but problems are still severe. For results see Figure 1 of Figshare.
>
> **Action** We added GPT-OSS 120B to our evaluation results and updated relevant figures and discussions.
>
> **W2: Should include closed-source judges or narrow the claim.**
>
> **A2:** Thank you for this suggestion. We agree that explicitly evaluating frontier proprietary models as judges would better support our broad claim. Our supplementary data already includes GPT-5 judge results, which follow the same trend as the open-source models and offer no meaningful improvement. We initially did not add these results to our paper because of the reasons givens in the manuscript (but the data is available to the reviewer if they are interested in exploring). To provide a more comprehensive picture, we also ran experiments using Gemini and Claude as judges across all model-attack combinations, covering a total of 1100 samples (50 per attack-model combination). We use the prompt provided in the StrongReject paper for judging with proprietary models, which also served as the basis for our human rubrics. Results are largely consistent with the failure pattern we observe across open-source judges. However, closed-source judges perform slightly better. Yet, Claude and Gemini refuse to judge in 8.27%, and 1.0% of cases, respectively, in our experiments. In the case of GPT-5, we were even banned from using their API after judging ~2.6k samples, due to the toxic nature of the examples. Beyond reproducibility issues, this adds further unreliability when using proprietary models as judges.
>
> For new results, please see Figure 1 and Figure 2 of the Figshare document.
>
> **Action** We will incorporate these results into the main paper to better support our claims.
>
> **Minor:** Thank you for the careful reading and attention to detail, the typographical error has been corrected.

---

> > ### Author Rebuttal · Reviewer_p2t1 · 2026-04-04
> >
> > My concerns have been addressed, so I will increase my score to 4.

---

> > > ### Author Response · Authors · 2026-04-07
> > >
> > > Thanks for the constructive feedback! We are glad we were able to resolve the relevant concerns.

---

### Official Review · Reviewer_s5Pc · 2026-03-11

**Soundness:** 3
**Presentation:** 3
**Significance:** 2
**Originality:** 2
**Overall Recommendation:** 4
**Confidence:** 3

**Summary:**

This paper examines how effectively an assortment of popular LLM judges can evaluate the success of adversarial attacks. By comparing results of the judges to manually annotated data, the authors see how different shifts in the distribution (models, attacks) affect the performance of these judges. They find that most judges do not perform better than random chance and do not closely align with human ratings, and they further identify many failures modes with these judges. Some additional strategies for making judges have more reliable annotations, such as using judge precision are provided, along with 2 benchmark evaluation sets.

**Compliance With Llm Reviewing Policy:**

Affirmed.

**Final Justification:**

I believe the discussion in the rebuttal about the contributions of the work was helpful in differentiating the key findings from prior works and specifically highlights what the key takeaways of the work are (this content should be included in the final revision). As my main concern on limited novelty compared to prior works is mostly addressed I am willing to increase my score to a 4.

**Key Questions For Authors:**

What prompts were used for the judges and what were the attack settings used for each attack?

**Limitations:**

Yes

**Strengths And Weaknesses:**

**Strengths**

The paper showcases key issues with using established LLM judges to evaluate adversarial robustness, and provides many  experiments that cover different aspects of distribution shift. The structure of the paper is clear and easy to follow. Benchmark data is provided for future work.

**Weaknesses**

1. My main concern is with limited novelty. As authors state prior works like StrongReject have shown the failure modes of LLM judges and how adversarial attack success is overinflated. Additionally many works such as [1] have shown issues with fine-tuned judges on in terms of generalizability and other factors around LLM like how they take short cuts [2]. The main thing this paper mainly does more evaluation on the effect of distribution shifts but this does not provide major insights and merely elaborates on prior work. I think to improve the paper it would be beneficial to see whether based on these results trying to align the judges to the human annotations and seeing whether the issues persist would be very beneficial to see how well this issue can be rectified.

2. The analysis is only conducted on one dataset so the generalizability of the results is in question.

3. Missing implementation details like prompts used, adversarial attack optimization details, how many people were used to annotate each example and agreement between the human evaluators (and not just those from Labelbox) etc.

[1] [An Empirical Study of LLM-as-a-Judge for LLM Evaluation: Fine-tuned Judge Model is not a General Substitute for GPT-4](https://aclanthology.org/2025.findings-acl.306/) (Huang et al., Findings 2025)

[2] Oriyad, Arash Mari et al. “The Silent Judge: Unacknowledged Shortcut Bias in LLM-as-a-Judge.” ArXiv abs/2509.26072 (2025)

---

> ### Author Rebuttal · Authors · 2026-03-29
>
> We highly appreciate the constructive review! New results can be found here: https://figshare.com/s/0e60460a72506d06ca65
>
> **W1: Limited novelty. Would aligning judges to human annotations resolve the issue?**
>
> **A1:** We acknowledge that several studies on judge reliability exist, and we welcome the opportunity to clearly distinguish our contributions from prior work such as StrongREJECT. Below, we summarize the aspects that are unique to our study.
>
> 1. **Considerably different message concerning reliability** Crucially, we find that judges are considerably worse than previously stated for adversarial evaluations, even compared to those papers criticizing the reliability of judges (no paper reports performance close to random guessing). We believe a surprising finding that overturns established assumptions is itself a meaningful contribution.
>
> 2. **Evaluation under realistic distribution shift** StrongREJECT and related work do not investigate whether their improvements transfer to realistic evaluation conditions in academia and industry. Namely, evaluations under attack shift, model shift, and data shift. We show that the *interaction* of these shifts is what causes reliability to collapse.
>
> 3. **Impact on scientific findings.** Beyond measuring evaluator accuracy, the consequences of judge failures for the scientific claims that depend on them. We show that judge errors can change which attack appears most effective, meaning that comparative analyses of attacks and defenses may be artifacts of judge unreliability.
>
> 4. **Other findings.** We find judge ensembling to be highly ineffective in realistic evaluation conditions (a finding opposite to the one in JailbreakJudge), or that attack optimization does not improve harmfulness if measured by humans for all tested attacks.
>
> **Action:** We agree that, in the current version of the paper, it was difficult to identify aspects unique to our study. We now emphasize the unique aspects of our study in the introduction of the paper and thank the reviewer for their comment.
>
> Regarding alignment with human annotations, we kindly refer to reviewer **Vcwy**, **A3.1**, who had a similar concern.
>
> **W2: Generalizability: analysis conducted on only one dataset.**
>
> **A2:** While our experiments use HarmBench, we note that it is a highly diverse benchmark spanning a broad range of harmful behaviors across multiple semantic categories. We observe substantial variation in judge performance *across* these behaviors for all attacks and models, demonstrating that the dataset already captures meaningful heterogeneity in evaluation difficulty; it is not a narrow or homogeneous test set.
>
> Maybe more importantly, HarmBench is one of the most widely adopted benchmarks for adversarial robustness evaluation, making it precisely the setting in which our findings are most consequential for the community. Alternatives like JBB have considerable overlap to HarmBench and would not add further diversity. Lastly, annotating more data would considerably increase cost.
>
> **Q1: Prompts used for judges; annotation details; attack hyperparameters.**
>
> **A (Q1):** This is a valid point, and we agree that including these details will improve reproducibility. We have added all of them to the paper: annotation details in the main text and judge prompts and attack hyperparameters in the appendix.
>
> ### Attack Hyperparameters
>
> See page 2 of Figshare link
>
> ### Annotation Details
>
> We follow the single-annotator per sample recommendation from [A], supported by preliminary evidence of high inter-rater agreement. Quality was ensured through a four-stage protocol:
>
> 1. All labelers (including those hired through Labelbox) were provided with a detailed annotation guide that they were required to read prior to labeling. The guide has been added to the appendix (external text sources not allowed during rebuttal).
>
> 2. We validated inter-rater agreement on 50 randomly selected samples from the 2,370 we labeled in-house, obtaining an agreement of 0.82. Combined with the theoretical justification in [A], this supports the single-annotator approach for the Labelbox-collected samples.
>
> 3. Labelbox annotators were required to pass a qualification quiz of 10 pre-labeled examples before accessing the annotation task.
>
> 4. As an additional quality check, we manually reviewed 90 Labelbox assignments and found 95% agreement with our own labels on the harmful/benign distinction. We attribute the higher rater agreement with labelbox raters to the quiz they had to conduct before labeling.
>
> [A] Khetan et al., "Learning From Noisy Singly-Labeled Data," ICLR 2018.
>
> ### Judge Prompts
>
> We agree that adding this information directly to the paper, makes it easier to reproduce our work. We use the setup implemented in judgezoo (https://github.com/LLM-QC/judgezoo), which is based on the paper implementation and do not conduct further changes. We added the description of judge prompts to the paper (see page 3 onward in Figshare).

---

> > ### Author Rebuttal · Reviewer_s5Pc · 2026-04-03
> >
> > Thank you for the response and providing more details about implementation details and clarifying the novelty. I still have some concerns though.
> >
> > Although the work does acknowledge some additional insights compared to prior work, I still see some issues. Namely,  although you state that no previous works show judge performance is akin to random guessing, because the evaluation is so limited based on only Harmbench (especially since previous works cover other datasets and scenarios which led to their differing conclusions) it is difficult to see whether these findings generalize or are just artifacts from using Harmbench. Even though the dataset is popular and relatively diverse, it has it's own set of biases and limitations and is not fully representative of real world data and I still believe that validating the analysis on a different set of data even in a limited capacity is necessary. There are many more safety datasets like Cosafe or Aegis that would strengthen the claims made.
> >
> > Furthermore, may I ask why you believe the findings overturn established assumptions? Just saying that judges perform worse in the specific cases you look at does not seriously overturn most of the previous work, but elaborates on it. How would it cause a shift in how the community uses LLM judges? Is the conclusion to not use them at all for evaluating attack performance or just in very specific scenarios?

---

> > > ### Author Response · Authors · 2026-04-03
> > >
> > > Thank you for the constructive discussion.
> > >
> > > ## On the choice of HarmBench:
> > >
> > > We want to address a practical constraint that motivates our dataset choice. Collecting sufficient human labels to make robust claims across every victim model and attack combination is already expensive at scale; labeling effort grows considerably with each additional axis of variation. Given this, we faced a concrete trade-off: label small subsamples across many datasets, or allocate our annotation budget to one well-established benchmark.
> > >
> > > We chose the latter for two reasons. First, HarmBench is one of the most widely used benchmarks in the adversarial robustness literature, meaning our labeled data (and the resulting ReliableBench subset) can be reused by the community without additional annotation cost whenever HarmBench is used, a property that would not hold if we had spread labels across less common datasets. Second, fragmenting the annotation budget across datasets would have left each subset too small to support the stratified analyses (across behaviors, attacks, and models) that are central to our findings.
> > >
> > > We also see no evidence that our findings are specific to HarmBench. If dataset-specific idiosyncrasies were driving our results, we would expect a consistent bias across behaviors. Instead, Figure 4c shows substantial heterogeneity: some behaviors are consistently misjudged across all judges, others show high variance, and others are reliably judged well. Combined with the consistency of our findings across 7 judge models, including 3 proprietary ones, this suggests the observed effects reflect general properties of judge reliability under distribution shift rather than artifacts of the dataset. We agree that replicating these findings on additional datasets would be valuable, and we believe our work and the human-labeled data we release provide a strong incentive for the community to invest in this direction and are therefore valuable to the overall safety community, but we believe it's unreasonable for a single group of researchers to conduct a study of this scale.
> > >
> > > ## On whether findings overturn established assumptions:
> > >
> > > This is a valid question, and we are happy to elaborate on our reasoning. The implicit takeaway from prior work is that judges are imperfect but usable, and that ensembling or calibration can mitigate issues. For example, StrongReject reports 0.819 correlation for the HarmBench judge and 0.900 for the proposed StrongReject. A relevant improvement, but HarmBench looks reasonably accurate in their tests. Similarly, JailJudge reports accuracies over 80% for 10/15 judges on JBB (which has a strong overlap with HarmBench). Their results imply that one could not use these new judges in their own safety study and would very likely end up with the same scientific takeaways, with a bit more noise. Our contribution is to show that this picture changes substantially under realistic deployment conditions, to the point where one can question the use of judges in these scenarios altogether, as we will elaborate on in the next point.
> > >
> > > Regarding the practical implication:
> > >
> > > i) Our results indicate that small and medium-sized models, such as Llama-8B-IT or even Gemma-27-B, should not be used for adversarial evaluations that rely on judges without considerable effort spent in manually verifying claims. Yet, models of this size are among the most widely used in safety evaluations, both for training defenses and for evaluating attacks (e.g., [1,2,3,4,5]).
> > >
> > > ii) Comparative claims about attack effectiveness and defense robustness should be treated with greater skepticism when judge reliability under the relevant distribution shift has not been verified. This has not been reported in prior work.
> > >
> > > iii) We provide concrete tools for measuring robustness more reliably (ReliableBench) and for testing whether new judges are likely to perform better in these evaluation conditions (JudgeStressTest).
> > >
> > > The implications of prior work and ours are considerably different, and we believe these findings are crucial to the safety community that relies on benchmarks like harmbench and evaluations on LLama models to give a reliable signal for research progress.
> > >
> > > **Actions:** We provided concrete suggestions to authors on how to interpret our findings and design safety evaluations to achieve reliable results, based on the response above.
> > >
> > > [1] S. Xhonneux et al., "Efficient Adversarial Training in LLMs with Continuous Attacks", NeurIPS, 2024
> > >
> > > [2] A. Sheshadri et al., "Latent Adversarial Training Improves Robustness to Persistent Harmful Behaviors in LLMs", TMLR, 2025
> > >
> > > [3] P. Chao, et al., "Jailbreaking Black Box Large Language Models in Twenty Queries", NeurIPS, 2023
> > >
> > > [4] A. Zhou et al., "Universal and Transferable Adversarial Attacks on Aligned Language Models", ArXiv, 2023
> > >
> > > [5] M. Mazeika, et al., "HarmBench: A Standardized Evaluation Framework for Automated Red Teaming and Robust Refusal", ACM, 2024

---

### Official Review · Reviewer_Vcwy · 2026-03-14

**Soundness:** 3
**Presentation:** 3
**Significance:** 3
**Originality:** 3
**Overall Recommendation:** 4
**Confidence:** 4

**Summary:**

This work studies the accuracy of LLM judges on jailbreaking responses across several domain shifts such as model shift, attack shift, and data shift. To do this, they collect generated attacks from 5 attacks on 100 tasks from Harmbench, and the corresponding responses from the target models. The main finding is the judges' labels are not widely consistent with the human label on ~6k data, and that the results of many attacks are inflated using this inconsistency and the randomness in the judge.

**Compliance With Llm Reviewing Policy:**

Affirmed.

**Key Questions For Authors:**

Please see above.

**Limitations:**

Please see above.

**Strengths And Weaknesses:**

**Strengths**:

1- This is a thorough study on par with previous studies such as StrongReject that aim to measure the accuracy of the existing judges in the literature. There needs to be more efforts in this direction in order to calibrate the results of the jailbreaking methods. For instance, on of the findings is that BoN's result is largely inflated due to this effect.

2- The paper is really well written and I could grasp all of the ideas and the importance of the contributions at the first glance.

3- They have conducted a significant human label collection in order to do their analysis, mostly missing in previous studies.

4- There are several other important observations: (i) the judges are less accurate on smaller models (perhaps because the answers are less fluent); or (ii) judges agreement does not correlate with the accuracy. This is important since many other papers attempt to avoid the randomness of the judge using agreement methods.

5- They propose a benchmark in this regard to evaluate the judges on the jailbreakings.

**Weaknesses**:

1- I think the main missing comparison is with StrongReject. StrongReject also collects human labels to measure the effectiveness of several judges in a broad study. It should be stated what is new in this work that was missing in previous work.

2 - Moreover, as the StrongReject paper explains, they use the multiplication of a "compliance rate" and "response helpfulness" as the final score. This is because in many cases the model might comply but the response is a mere hallucination and cannot inflict harm. This factor is missing in this paper and only focus on the compliance as described in Section 3.5.

3- A missing piece to me is using the collected data to build better judges here as previous work does, and evaluate a larger corpus of attacking methods.

---

> ### Author Rebuttal · Authors · 2026-03-29
>
> We thank the Reviewer for the constructive and helpful review. New results can be found here: https://figshare.com/s/0e60460a72506d06ca65
>
> **W1: Missing a detailed comparison to StrongReject**
>
> **A1:** We agree that there are several studies on judge reliability, and we appreciate the opportunity to carefully distinguish our contributions from prior work such as StrongREJECT. Here, we summarize aspects unique to our study.
>
> 1. **Considerably different message concerning reliability** Crucially, we find that judges are considerably worse than previously stated for adversarial evaluations, even compared to those papers criticizing the reliability of judges. We give a new perspective on the extent of the unreliability. Prior evaluator papers, including StrongREJECT, consistently report high human agreement even for the *baseline* methods they aim to improve, whereas we find that all judges in our study are fundamentally unreliable (under realistic evaluation conditions, see next).
>
> 2. **Evaluation under realistic distribution shift** StrongREJECT and related work do not investigate whether their improvements transfer to realistic evaluation conditions in academia and industry. Namely, evaluations under attack shift, model shift, and data shift. We show that the *interaction* of these shifts, which is unavoidable in standard red-teaming, is what causes reliability to collapse.
>
> 3. **Downstream impact on scientific attribution.** Beyond measuring evaluator accuracy, the consequences of judge failures for the scientific claims that depend on them. We show that judge errors can change *which attack appears most effective*, meaning that comparative analyses of attacks and defenses may be artifacts of judge unreliability rather than reflections of genuine safety properties. StrongREJECT does not examine this attribution problem.
>
> 4. **Other findings.** We will also highlight the unique findings mentioned by the reviewer, such as the correlation between accuracy and judge agreement, or that judges are less accurate on smaller models.
>
> **Action:** We have added a more detailed comparison with StrongREJECT to the related work section of the paper, explicitly discussing the differences in scope, methodology, and findings.
>
> **W2: StrongReject uses compliance rate and response helpfulness. This factor is missing in the paper.**
>
> **A2:** Thank you for raising this point. We do calculate harmfulness using the StrongREJECT formulation (as a combination of compliance rate and response helpfulness), consistent with the original paper. This was not made sufficiently clear in the submission.
>
> **Action:** We now provide a comprehensive description of all judge configurations in the paper rather than referring solely to the implementation details in Judgezoo and the original works. See also page 3 onward of the figshare document.
>
> Moreover, we added the exact formula on how scores are calculated for every judge to our work. E.g., for StrongReject we use the formula given in their repository: https://github.com/dsbowen/strong_reject/blob/main/strong_reject/evaluate.py
>
> **W3: More attacking methods and training a judge on the collected data.**
>
> **A3:** We agree that additional attack methods would strengthen the study. Unfortunately, each method adds significant labeling overhead, as annotations must be collected for every attack–model combination. If the reviewer has a specific method in mind, we are open to conducting targeted additional labeling for the camera-ready version for one or two attacks for all models.
>
> **A3.1** Regarding training a judge on our data, we had two substantive reasons for not doing so.
>
> First, our results demonstrate that reliability failures are not architecture-specific. They persist across a diverse set of judges (AegisGuard, the HarmBench classifier, JailJudge, and LlamaGuard-3) and are not resolved by scaling training data or budget. AegisGuard was trained on 26,000 human-annotated examples and still exhibits the same failure modes. We additionally ran experiments using Claude, and Gemini as judges and observed the same degradation, suggesting that training a better judge on more data is insufficient to achieve high rating accuracy in the combinatorial space of attacks and victim models (see new results in **p2t1 W2**).
>
> Second, training a judge on our own benchmark data would introduce a conflict of interest. As the authors who identified these failure modes, optimizing for our own benchmark would weaken the independence of the evaluation and risk overstating progress.

---

> > ### Author Rebuttal · Reviewer_Vcwy · 2026-04-04
> >
> > I appreciate the answers to my questions. I have another question: what is the main difference between your data collection method and previous data collection for building judges such as StrongReject? More specifically, what is your advantage?
> >
> > I can only see the false negative results for StrongReject. What about false positives? How biased is StrongReject overall (i.e., what would the final error in the reported ASR)
> >
> > Can you also further discuss the reason for this bias in the judges? what are the possible reasons for the bias?

---

> > > ### Author Response · Authors · 2026-04-04
> > >
> > > Thank you for the constructive engagement!
> > >
> > > **Q1: Difference in data collection**
> > >
> > > There are several key differences between our data collection methodology and that of StrongReject.
> > >
> > > **1. Prefiltering and annotation strategy.** The most fundamental difference in data collection is our prefiltering step. In the StrongReject paper, no prefiltering is applied before human annotation and data was sampled for labeling uniformly. From our own experiments, we observed that the vast majority of attacks fail: the model simply refuses or produces irrelevant content, which is simple to judge (see also Figure 3 in the Figshare). Without prefiltering, the annotation budget is overwhelmingly spent on obvious true negatives that are uninformative for studying judge reliability. We instead apply an automatic prefilter (StrongReject classifier, H > 0.5) to restrict human annotation to judge-positive samples, where failures are more informative. This yields a roughly balanced partition of true positives and true negatives (Section 3.6). Note that without filtering, metrics like accuracy or mean absolute error will be flawed, as a judge that labels everything as a negative will achieve good performance.
> > >
> > > Moreover, we provide a dataset of 6,642 human-labeled samples (vs. StrongReject's 1,361). Crucially, this scale is what makes the combinatorial analysis across attacks, models, and behaviors possible with sufficient samples per condition (at least 105 per attack-model pair, at least 785 per model). For comparison, as reported in StrongReject's Appendix D, only 9 unique jailbreaks were labeled for Llama-2 and 4 for GPT-4, which is insufficient for a reliable analysis. For more details regarding annotation (see Reviewer **s5Pc, A (Q1)**).
> > >
> > > **2. Focus on strong adversarial attacks.** Another important difference lies in the types of attacks evaluated. StrongReject's 17 jailbreak methods are template or encoding-based (Base64, ROT-13, AIM, translation attacks, disemvowel, etc.), static transformations that are applied to prompts without any feedback from victim model or judge. By contrast, our attack selection is specifically designed to strong attacks and methods with varying degrees of judge interaction: GCG-REINFORCE *explicitly* uses judge signal in its optimization loop, BoN *implicitly* exploits judge weaknesses through extensive sampling (up to 10,000 trials per query), and GCG and PAIR iteratively refine prompts using judge feedback. This design directly enables us to study judge hacking. For example, we show that sampling-based methods like BoN amplify false positive rates (Figure 1, Figure 8b). Moreover, it is more relevant for actual evaluations, that mostly use strong adversarial attacks like GCG.
> > >
> > > Importantly, the limited amount of data combined with the large number of adversarial attacks does not allow for a thorough study of the interaction of data, model, and attack shift in StrongReject, which we specifically analyze in our work.
> > >
> > > **Q2: What about StrongReject FP? How biased is StrongReject overall?**
> > >
> > > We provide an analysis regarding StrongReject false positives in Section 3.6. Here, we find that 78% of all positives from StrongReject are false positives. Regarding false negatives, we added a new analysis that we describe in our answer to Reviewer **wwTo, A1**. Annotating 20 samples from each attack for StrongReject and Llama-8B-IT reveals a false negative rate of close to 0 for all attacks. Thus, the effect of false positives is considerably greater on the final ASR and considering false negatives as well does not change the results reported in the paper.
> > >
> > > We initially did not include StrongReject in the main judge comparison study because we also used it for filtering, and we wanted to avoid introducing a circular bias.
> > >
> > > **Action:** We will add a separate section to the appendix that summarizes statistics of the StrongReject classifier in a dedicated table.
> > >
> > > **Q3: Reasons for the bias?**
> > >
> > > A manual analysis of cases, where the majority of judges disagree with the human assessment reveals several failure cases:
> > >
> > > **1. Reliance on surface-level linguistic patterns.** This was already noted in StrongReject but we find it to remain true for all existing judges. Seemingly harmful text with no semantic meaning consistently triggers harmful ratings from all judges.
> > >
> > > **2. Judges fail out of distribution.** Judges often assign harmful scores to nonsense outputs that are forced by the adversarial attacks. For example, repetition of harmful or even nonesense words words.
> > >
> > > **3. Translation.** Judges often assign toxic scores to outputs in non-English languages independent of their content.
> > >
> > > **4. Harmful but unrelated.** Victim models sometimes generate toxic semantically meaningful answers that are not related to the question.
> > >
> > > **Action:** We added example failures to the paper and will release a React-based dataset explorer app to make exploring failure cases easier. We used this app to find these examples.

---

### Decision · Program_Chairs · 2026-04-30

**Decision:**

Accept (regular)

**Comment:**

The recommendation is based on the reviewers' comments, the area chair's evaluation, and the author-reviewer discussion. This paper studies the reliability issues of LLM-as-a-judge in safety/robustness assessment. All reviewers find the studied setting novel and the results provide new insights. The authors’ rebuttal has successfully addressed the major concerns of reviewers. In the post-rebuttal phase, all reviewers were satisfied with the authors’ responses and agreed on the decision of acceptance.